# S-Quant: Rethinking Weight Quantization with Seed-Based Generation

**Mingzi Wang**[1]  **Lancheng Zou**[1]  **Shuo Yin**[1]  **Zhuolun He**[1]  **Bei Yu**[1]

## Abstract

The progressive scaling of large language models (LLMs) has consistently enhanced multimodal understanding and advanced reasoning capabilities, but has substantially increased computational and hardware execution overhead. In this paper, we present S-Quant, a novel post-method that compresses only model weights. We partition each weight tensor into fixed-size blocks and assign a single seed to each block. The seed drives a hardware-friendly Linear Feedback Shift Register (LFSR) generator that dynamically produces multiple basis matrices. Each block is then reconstructed as a linear combination of these basis matrices, with block-specific coefficients, which substantially reduces the amount of stored data, increases the data-transfer efficiency between memory and compute units, and consequently speeds up memory-bound inference for large language models. Experimental results on different LLM models ranging from 7B–70B parameters show that S-Quant attains state-of-the-art performance when weights are compressed to approximately 3-bit or 4-bit. We also design a dedicated ASIC accelerator that achieves a 4× speed-up for memory-bound LLM inference. Code is available at https://github.com/wmz-max/S-Quant.

## 1. Introduction

Large language models (LLMs) deliver state-of-the-art results across a wide range of natural language processing tasks, and their strong language understanding has been extended successfully to multimodal problems (Touvron et al., 2023; Zhang et al., 2022). However, their large computational and memory footprints remain a major barrier to practical use. For example, GPT-3 (Brown et al., 2020), with roughly 175 billion parameters, requires on the order of 350 GB of memory when stored in FP16, which effectively translates into the need for at least five NVIDIA A100 80GB GPUs to perform inference. The resulting computation and inter-GPU communication overheads make real-world deployment costly and technically challenging.

Autoregressive LLM inference is primarily constrained by memory bandwidth, as the retrieval of weight and activations dominates execution time. Off-chip DRAM accesses entail orders-of-magnitude greater latency and energy consumption compared to on-chip multiply–accumulate (MAC) operations. Consequently, reducing the memory footprint through model compression represents the most effective strategy for mitigating inference latency, lowering power consumption, and reducing deployment costs. Quantization (Xiao et al., 2023) represents model weights and activations using lower-precision formats to decrease storage and bandwidth demands. Pruning (Ma et al., 2023; Sun et al., 2023) removes parameters judged to be redundant, reducing model size and computational cost. However, most post-training compression methods rely on calibration data and suffer severe accuracy degradation under extreme compression. So, we explore one question whether a calibration-free compression method can be designed that maintains acceptable accuracy at extreme compression?

We present S-Quant, a weight-only compression method that achieves extreme compression—approximately 3-bit effective precision—while maintaining acceptable accuracy. S-Quant partitions each weight matrix into fixed-size blocks and approximates each block with a small set of basis tensors that we deterministically generate from a seed. We reconstruct a block by linearly combining the generated basis tensors and multiplying each basis by an optimal coefficient. As a result, we represent and transmit a block's parameters by a single seed and its coefficient vector, which substantially reduces storage and communication bandwidth compared with storing raw weights. In contrast to a previous work SeedLM (Shafipour et al., 2024), which reconstructs each block using floating-point matrix multiplications and incurs significant hardware overhead, S-Quant avoids expensive matrix operations. As a result, S-Quant achieves higher accuracy while substantially reducing hardware cost.

S-Quant has two practical challenges. First, weight blocks exhibit large numerical variability, so we must determine

---

[1]The Chinese University of Hong Kong, China. Correspondence to: Bei Yu <byu@cse.cuhk.edu.hk>.

*Proceedings of the $43^{rd}$ International Conference on Machine Learning*, Seoul, South Korea. PMLR 306, 2026. Copyright 2026 by the author(s).

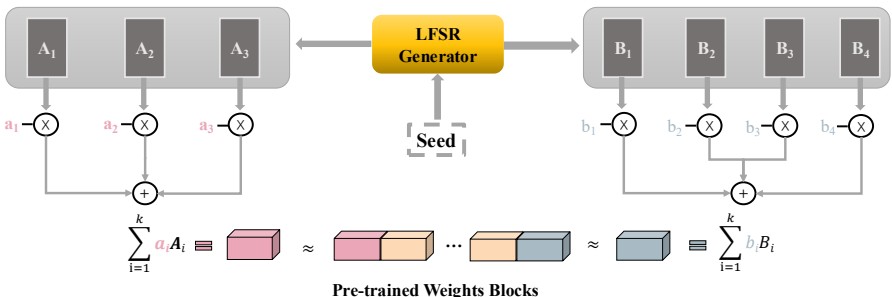

*Figure 1.* S-Quant Framework.

how many basis tensors to generate for each block. Second, design parameters such as block size and seed length trade off against post-compression accuracy and average bits per weight, so we must find a Pareto balance among these objectives. To address the first challenge, we define the explained energy ratio to measure the fraction of a block's energy retained by a given basis set, and we use this metric to adaptively select the number of basis tensors per block. To tackle the second challenge, we formulate the selection of block size, seed length, and associated hyperparameters as a multi-objective design-space exploration problem. We then employ Bayesian optimization to identify operating points that achieve a desirable trade-off among accuracy, compression rate.

S-Quant increases on-chip computation within bounded limits to reduce off-chip memory accesses and improve effective chip-to-chip bandwidth, enabling extreme compression in multi-chip deployments. The tensor generator uses a linear-feedback shift register (LFSR), a communications-domain primitive (Win & Kyaw, 2008) that relies primarily on hardware-friendly shift and XOR operations. This choice yields a compact, deterministic, and easily pipelined hardware implementation.

We make the following contributions in this paper:

- We propose a novel weight-only compression method, S-Quant, which dynamically reconstructs each weight block from a seed using a Linear-Feedback Shift Register (LFSR) generator, substantially reducing stored information and increasing effective memory bandwidth.

- We adapt the number of basis tensors per block based on each block's explained energy and reconstruction error to balance compression and accuracy.

- We introduce an search strategy that employs Bayesian optimization to find optimal configuration parameters such as block size and seed length.

- We designed a custom hardware accelerator, imple-

mented in SystemVerilog, and demonstrated in simulation that for memory-bound LLM inference it achieves up to a 4× speedup.

- Extensive experiments on LLM models ranging from 7B–70B parameters show that S-Quant attains state-of-the-art performance with weights compressed to approximately 3-bit or 4-bit.

**Conflict of Interest Disclosure**  The authors declare that there are no financial conflicts of interest related to this work.

## 2. Preliminaries

### 2.1. LLM Weight Compression

Weight-only quantization targets representing model weights at reduced bit widths to lower storage and compute requirements. For example, GPTQ (Frantar et al., 2022) uses block-wise reconstruction to attain 3–4 bit quantization. SpQR (Dettmers et al., 2023b), OWQ (Lee et al., 2024), and AWQ (Lin et al., 2024) prioritize weights associated with large-magnitude activations. Consequently, SpQR and OWQ adopt mixed-precision schemes to preserve those critical weights, while AWQ applies channel-wise scaling to avoid the hardware inefficiencies of mixed precision. Qlora (Dettmers et al., 2023a) recovers performance by performing parameter-efficient fine-tuning on the quantized model. In QuIP# (Tseng et al., 2024), Hessian analysis of calibration data helps make rounding decisions during quantization.

LLM pruning has emerged as a critical challenge as large language models continue to scale in size. Conventional pruning techniques, which typically involve retraining the entire model, are computationally expensive and increasingly infeasible for models of this magnitude. Recent work has shifted toward post-training pruning approaches (Frantar & Alistarh, 2023; Sun et al., 2023; Das et al., 2023), where specialized scoring functions are employed to assess the significance of weights and prune less influential

components without requiring costly retraining. In addition, SliceGPT (Ashkboos et al., 2024) advances structured pruning by eliminating rows or columns of weight matrices according to eigenvectors and eigenvalues derived from the input, thereby offering a more principled strategy for reducing model complexity.

Recent work shows that network weights can be compactly represented by a pseudo-random generator seed together with compact coefficient vectors. PRANC (Nooralinejad et al., 2023) compresses entire networks by orders of magnitude to reduce storage and improve transmission efficiency. LoRA (Hu et al., 2022) lowers weight storage by injecting trainable low-rank decomposition matrices into each layer. NOLA (Koohpayegani et al., 2023) builds on LoRA by expressing low-rank factors as linear combinations of random basis vectors, further reducing memory footprint and computational overhead. SeedLM (Shafipour et al., 2024) is first use pseudo-random generator in LLM weight compression, but block reconstruction relies on floating-point multiplications between basis tensors and their coefficients, significantly increasing power consumption and silicon area.

## 2.2. Linear Feedback Shift Register

A linear feedback shift register (LFSR) is a compact and low-cost mechanism for generating pseudo-random binary sequences and has been widely used in digital circuits and embedded systems. Rather than relying on complex arithmetic operations, LFSRs are designed around simple and deterministic state update rules, which enable efficient sequence generation while keeping both hardware and memory overhead low.

An LFSR consists of a binary register of length $K$. At iteration $t$, its internal state can be represented as $\mathbf{s}_t = (s_t^{(1)}, s_t^{(2)}, \ldots, s_t^{(K)})$, where each component corresponds to a single bit in the register. At any time, the system evolves solely based on the current state, without accessing past states or auxiliary storage structures.

The state evolution can be expressed in a compact form as

$$\mathbf{s}_{t+1} = \mathbf{T}\,\mathbf{s}_t, \tag{1}$$

where $\mathbf{T}$ denotes a fixed state transition operator. This operator is not intended for numerical matrix computation; instead, it serves as a concise representation of the update rules governing the register. In particular, $\mathbf{T}$ jointly encodes how the register contents are shifted and how the feedback bit is generated from selected state positions.

From an implementation perspective, the update process involves two basic types of operations. Most bits are simply shifted to adjacent positions at each iteration, while a new feedback bit is injected at one end of the register. The feedback bit is computed as the XOR of a predefined

subset of the current state bits. Since the update relies exclusively on shift operations and bit-wise XORs, LFSRs can be implemented without multipliers, complex control units, or elaborate combinational logic, resulting in very low hardware complexity and energy consumption.

From a memory standpoint, LFSRs are equally efficient. Only the current $K$-bit state needs to be stored in order to generate pseudo-random sequences of arbitrary length on demand. This approach avoids explicit storage of full random sequences, masks, or large lookup tables, and ensures that the memory cost remains constant regardless of the sequence length. Such a minimal state representation makes LFSRs particularly well suited for resource-constrained environments and implementations where storage efficiency is a primary concern.

## 2.3. Hardware Implementation Cost of Different Arithmetic Operations

In modern deep learning systems, weight computation is commonly formulated as floating-point matrix multiplication, where the dominant operation is floating-point multiply-accumulate (MAC). Prior studies have shown that, in area- and energy-constrained hardware implementations, such dense floating-point computations incur substantial cost in terms of silicon area and power consumption (Horowitz, 2014; Jouppi, 2017).

From the perspective of arithmetic unit design, different operations exhibit markedly different implementation costs. In particular, floating-point multipliers are significantly more complex than adders or simple accumulation units, requiring deeper logic pipelines and higher switching activity. When deployed at scale for dense matrix computations, the extensive use of floating-point multipliers leads to increased hardware complexity and elevated energy usage (Chen, 2016; Sze, 2017). In contrast, arithmetic patterns that reduce the number of floating-point multiplications and rely primarily on addition or accumulation are generally more favorable for efficient hardware realization.

Motivated by these considerations, many hardware-efficient designs aim to reformulate dense matrix computations into accumulation-dominant forms, where a small number of scalar scaling operations are followed by lightweight accumulation. Such designs reduce the reliance on costly floating-point multipliers, simplify data paths, and lower overall energy consumption, while remaining effective for neural network inference workloads (Han, 2016).

# 3. S-Quant

## 3.1. Seed-Based Weight Compression

Model parameters are represented using compact linear combinations of deterministic pseudo-random bases generated from a linear-feedback shift register (LFSR) with a fixed register length.

The integer-valued sequence generated from an LFSR initialized by seed $\sigma$ is written as:

$$Z(\sigma) \in \mathbb{Z}^T, \tag{2}$$

where $T$ denotes the sequence length.

Since the raw output of the LFSR is biased and uncentered, an affine transformation is applied to obtain a normalized sequence with controlled scale:

$$B(\sigma) = \frac{Z(\sigma) - 2^{R-1}\mathbf{e}}{2^{R-1} - 1}, \tag{3}$$

where $R$ is the register length and $\mathbf{e}$ denotes an all-ones vector of matching dimension.

The original parameter tensor is partitioned into $N$ non-overlapping blocks:

$$\Theta = [\,\theta^{(1)}, \theta^{(2)}, \ldots, \theta^{(N)}\,], \tag{4}$$

with each block associated with an independent seed $\sigma_n$.

From the normalized sequence corresponding to $\sigma_n$, a collection of $K$ consecutive basis elements is extracted:

$$\{B_1^{(n)}, B_2^{(n)}, \ldots, B_K^{(n)}\}, \tag{5}$$

which together span a low-dimensional subspace used for approximation.

Each parameter block is reconstructed as a linear combination of the corresponding basis elements:

$$\tilde{\theta}^{(n)} = \sum_{k=1}^{K} \alpha_{n,k} B_k^{(n)}, \tag{6}$$

where $\alpha_{n,k} \in \mathbb{R}$ are scalar coefficients obtained by solving a least-squares projection of $\theta^{(n)}$ onto the subspace spanned by the selected bases.

The reconstructed parameter tensor is formed by concatenating all reconstructed blocks:

$$\tilde{\Theta} = [\,\tilde{\theta}^{(1)}, \tilde{\theta}^{(2)}, \ldots, \tilde{\theta}^{(N)}\,], \tag{7}$$

For a fixed register length $R$ and feedback polynomial, a maximal-length LFSR cycles through $2^R - 1$ distinct internal states:

$$\mathcal{S} = \{s_1, s_2, \ldots, s_{2^R-1}\}, \tag{8}$$

allowing all corresponding output values to be precomputed and cached.

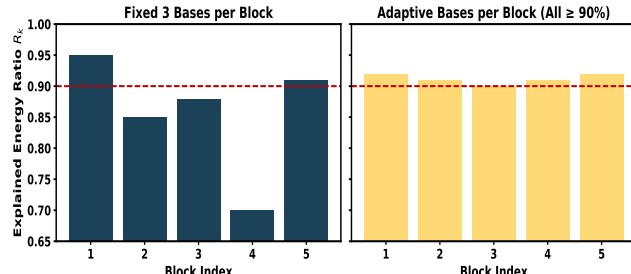

*Figure 2.* Comparison between Fixed and Adaptive methodologies(5 blocks selected in LLaMA 2-7B).

## 3.2. Explained Energy and Reconstruction Error

When a weight tensor is approximated using a limited set of basis tensors, a reconstruction error inevitably arises due to the projection onto a low-dimensional subspace. Let $T$ denote the original weight tensor (or a flattened block), and let $\hat{T}^{(k)}$ denote its approximation using $k$ selected bases. The reconstruction error is naturally measured by the squared Frobenius norm:

$$\mathcal{E}_k = \left\| T - \hat{T}^{(k)} \right\|_F^2, \tag{9}$$

which captures the energy of the residual orthogonal to the selected subspace.

To quantitatively assess how much of the original weight energy is preserved after compression, we define the explained energy ratio as:

$$R_k = \frac{\|P_k T\|_F^2}{\|T\|_F^2} = 1 - \frac{\|T - \hat{T}^{(k)}\|_F^2}{\|T\|_F^2}, \tag{10}$$

where $P_k$ denotes the orthogonal projection operator onto the $k$-dimensional subspace spanned by the selected bases. A higher $R_k$ indicates that most of the tensor energy is captured, corresponding to a lower reconstruction error.

This formulation is grounded in the orthogonal projection theorem in linear algebra, which guarantees that the best low-dimensional approximation is achieved via orthogonal projection and that the residual energy is orthogonal to the selected subspace (Strang, 2016; Trefethen & Bau III, 1997). Moreover, $R_k$ is conceptually analogous to the explained variance ratio commonly used in Principal Component Analysis (PCA) to evaluate how much of the original data variance is preserved in a low-dimensional embedding (Jolliffe, 2002; Golub & Van Loan, 2013).

## 3.3. Adaptive Basis Selection and Quantization

As shown in Figure 2 (left), using a fixed number of bases for all weight blocks leads to inefficient and uneven reconstruction quality. Some blocks are over-allocated, achieving energy ratios far above the target threshold $R_{\text{th}}$ and wasting bases, while other blocks fail to reach the threshold,

**Algorithm 1** Adaptive Basis Selection and Quantization

---

**Input:** weight blocks $\{w_i\}$, energy threshold $R_{\mathrm{th}}$, blocks per quantization group $G$

**Output:** selected seeds $\{s^*\}$, basis counts $\{k^*\}$, quantized coefficients $\{\hat{a}\}$

**for** each weight block $w_i$ **do**
    Initialize $k \leftarrow 2$
    Initialize $R_{\max} \leftarrow 0$
    **while** $R_{\max} < R_{\mathrm{th}}$ **do**
        **for** each candidate seed $s$ **do**
            Compute coefficients $a_i$ by Eq. (11)
            Compute explained energy ratio $R_k(s)$ using Eq. (10)
        **end for**
        Select $s^* \leftarrow \arg\max_s R_k(s)$
        Update $R_{\max} \leftarrow R_k(s^*)$
        **if** $R_{\max} < R_{\mathrm{th}}$ **then**
            $k \leftarrow k + 1$
        **end if**
    **end while**
    Record $s^*$, $k$, and $a_i$
**end for**
Group every $G$ consecutive blocks and quantize the corresponding coefficients $\{a_i\}$ to int8 using a shared scaling factor
**Return** $\{s^*\}$, $\{k^*\}$, $\{\hat{a}\}$

---

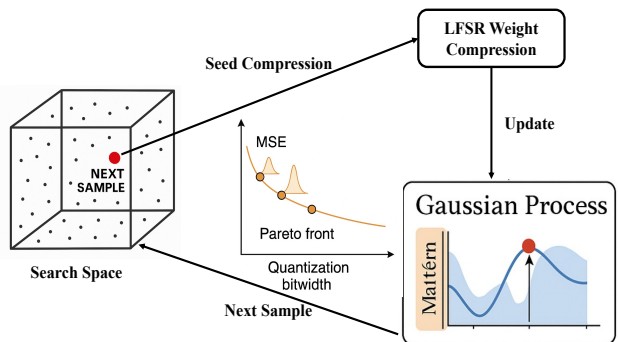

*Figure 3.* Searching Strategy.

resulting in poor reconstruction. To overcome this limitation, we adopt an adaptive strategy that dynamically selects the number of bases for each block to reach the target explained energy ratio (The ablation study about importance of adaptive basis selection is in **Appendix C**).

**Algorithm Overview.** For each weight block $w$, we start with a small number of bases ($k = 2$) and enumerate all LFSR seeds $s$. For each candidate seed, we construct the basis matrix $U_k(s) \in \mathbb{R}^{B \times k}$, whose columns are the $k$ normalized sub-blocks generated from seed $s$ according to Equation (3).

We compute the projection coefficients($a$) by solving the following least-squares problem:

$$a = \arg\min_{x \in \mathbb{R}^k} \|U_k(s)\, x - w\|_2^2, \qquad (11)$$

and evaluate the explained energy ratio using Equation (10).

If the best $R_k(s)$ is below the threshold $R_{\mathrm{th}}$, we increment $k$ and repeat the search process. Once the threshold is reached, we record the selected seed, basis count, and coefficients. Finally, coefficients of multiple blocks are grouped and quantized to int8 with a shared scaling factor. The detailed procedure is summarized in **Algorithm 1**.

Using this algorithm, as illustrated in Figure 2 (right), blocks

that are easier to approximate automatically use fewer bases, while harder blocks receive more bases. This ensures that all blocks reach the target $R_{\mathrm{th}}$ without redundant basis allocation.

### 3.4. Configuration Searching Strategy

Given a configuration, Algorithm 1 compresses the original weight tensor $W$ into $\hat{W}$ through adaptive basis selection followed by coefficient quantization. Each execution of the algorithm simultaneously induces two competing objectives. The first objective is the reconstruction error measured by mean squared error (MSE):

$$\mathcal{L}_{\mathrm{MSE}} = \frac{1}{N} \sum_{i=1}^{N} \|W_i - \hat{W}_i\|_F^2, \qquad (12)$$

where $W_i$ and $\hat{W}_i$ denote the original and reconstructed weight blocks, respectively. The second objective is the effective storage cost per block, expressed as:

$$Bitwidth = S + 8k + \frac{16}{G}, \qquad (13)$$

where $S$ is the length of the LFSR seed, $k$ is the number of adaptively selected bases, and $G$ is the number of blocks sharing a single FP16 scaling factor.

These two objectives are inherently conflicting: reducing storage cost by decreasing bitwidth typically increases reconstruction error, while allocating more bases improves accuracy at the expense of additional storage. Importantly, this trade-off is fully determined by a four-dimensional configuration:

$$T = \langle B,\, S,\, G,\, R_{\mathrm{th}} \rangle, \qquad (14)$$

where $B$ specifies the block size, $S$ controls the number of candidate basis sequences, $G$ determines the granularity of shared quantization scaling, and $R_{\mathrm{th}}$ defines the target explained energy ratio used in adaptive basis selection.

As a result, identifying an effective compression strategy can be naturally formulated as a multi-objective design

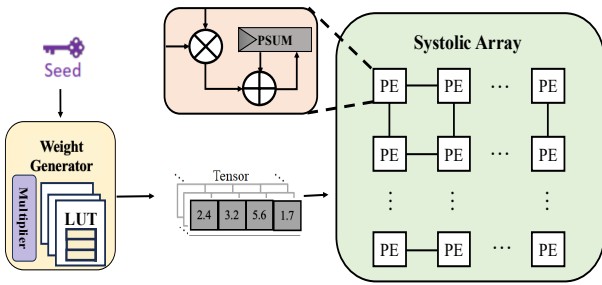

*Figure 4.* S-Quant Accelerator Implementation.

space exploration problem. Each configuration $T$ corresponds to a design point that yields a bi-objective outcome $\left(\mathcal{L}_{\mathrm{MSE}}(T),\ Bitwidth(T)\right)$, and the goal is to discover configurations that achieve a favorable balance between accuracy and storage efficiency. To efficiently explore this mixed discrete–continuous design space, we employ a Bayesian optimization framework, illustrated in Figure 3, with a Gaussian Process surrogate model using a Matérn kernel. At each iteration, the next configuration $T^*$ is selected by maximizing the Expected Incremental Predictive Volume (EIPV) (Shah & Ghahramani, 2016):

$$T^* = \arg\max_{T \in \mathcal{D}}\ \mathrm{EIPV}(T \mid \mathcal{D}), \qquad (15)$$

where $\mathcal{D}$ denotes the design space of all candidate configurations. The EIPV function is defined as:

$$\mathrm{EIPV}(T \mid \mathcal{D}) = \mathbb{E}_{\mathbf{f}(T)}\left[\Delta \mathrm{HV}\big(\mathbf{f}(T), \mathcal{P}\big)\right], \qquad (16)$$

with $\Delta\mathrm{HV}$ representing the hypervolume improvement obtained by augmenting the current Pareto set $\mathcal{P}$ with the candidate solution $\mathbf{f}(T)$. After the Pareto frontier is obtained, a single operating point is selected from the frontier as a Pareto-efficient solution that achieves a favorable compromise between reconstruction accuracy and storage efficiency, and the selected configuration is fixed and used in all subsequent experiments (The details of the derivation of BO and EIPV are in **Appendix D**).

### 3.5. Hardware Analysis

**Hardware Implementation.** We propose a seed-compression-aware accelerator because conventional GPUs, which are highly optimized for dense matrix-multiplication kernels, cannot efficiently reconstruct weights from compact seeds. We implement the Weight Generator and the Systolic Array in SystemVerilog and synthesize the designs with Synopsys Design Compiler (Ultra, 2017) targeting a 7 nm FinFET (Clark et al., 2016) standard-cell library to obtain area, timing, and power estimates for the hardware implementations. A cycle-accurate simulator is developed to evaluate end-to-end system performance, and CACTI (Muralimanohar et al., 2009) is employed to model on-chip

memory latency and power. The Weight Generator comprises a lookup table (LUT) that stores all LFSR states together with the multipliers required to reconstruct weights from seeds. The Systolic Array consists of processing elements (PEs), each integrating a multiplier, an adder, and a partial-sum buffer to accumulate intermediate results.

**Weight Reconstruction Pipeline.** Figure 4 depicts the end-to-end dataflow of the proposed accelerator. Seeds and coefficients are first loaded from memory into the Weight Generator. The generator uses the seed to rapidly index a lookup table (LUT) with single-cycle access to obtain the corresponding tensor, and reconstructs weights on the fly via multiplier and adder units. The reconstructed weights are subsequently streamed into the systolic array for efficient multiply–accumulate (MAC) operations (The actual overhead of reconstruction under different batch sizes is reported in **Appendix B**).

## 4. Experiments

### 4.1. Performance Analysis

We evaluate S-Quant using two complementary settings. Language modeling quality is measured via perplexity on WikiText-2 (Merity et al., 2016), and zero-shot accuracy is assessed on a collection of downstream tasks using the LM Evaluation Harness (Gao et al., 2021). We compare against SeedLM (Shafipour et al., 2024), AWQ (Lin et al., 2024), OmniQuant (Shao et al., 2023), and QuIP# (Tseng et al., 2024), relying on their official public implementations. SeedLM operates without calibration, while AWQ, OmniQuant, and QuIP# generally depend on layer-wise calibration with held-out data. S-Quant similarly requires no calibration and can be applied directly to pretrained checkpoints, yet achieves consistently lower perplexity across model scales and low-bit configurations. For AWQ and OmniQuant, we employ 4-bit integer quantization with channel-wise scaling to limit effective bitwidth growth (a group size of 128 adds roughly 0.25 bits per parameter). We do not fine-tune the quantized models for QuIP# or OmniQuant to maintain parity with calibration-free methods. Perplexity results on WikiText-2 are reported in Table 2, highlighting the trade-off between compression and accuracy. Zero-shot task performance is summarized in Table 4, where S-Quant matches or exceeds prior methods at the same bit budgets, demonstrating strong calibration-free robustness(The performance on Llama 2 13B, Llama 2 70B and Llama 3 8B is in **Appendix A**).

### 4.2. Hardware Analysis

In this work we perform hardware-level experiments to compare four weight-handling strategies—weights without compression, seed-compression (Shafipour et al., 2024), 4-bit weight quantization, and our S-Quant across Llama2 and

| Model | Method | Bits | ARC-Easy | ARC-Challenge | HellaSwag | WinoGrande | BoolQ | Mean |
|---|---|---|---|---|---|---|---|---|
| Llama 2 7B | Baseline | 16 | 74.58 | 46.33 | 75.98 | 69.06 | 77.74 | 68.74 |
| | **S-Quant** | **3.8** | **73.36** | **44.55** | **74.51** | **68.47** | **77.34** | **67.65** |
| | SeedLM | 4 | 73.23 | 44.54 | 74.45 | 68.43 | 77.19 | 67.57 |
| | AWQ | 4 | 70.58 | 43.94 | 74.96 | 68.75 | 78.29 | 67.30 |
| | QuIP# | 4 | 68.35 | 39.85 | 72.40 | 65.59 | 75.14 | 64.27 |
| | OmniQuant | 4 | 70.71 | 43.52 | 74.20 | 68.27 | 73.64 | 66.07 |
| | **S-Quant** | **2.7** | **70.01** | **41.39** | **70.74** | **66.38** | **74.29** | **64.56** |
| | SeedLM | 3 | 69.87 | 41.21 | 70.72 | 66.30 | 74.28 | 64.48 |
| | AWQ | 3 | 53.37 | 33.62 | 56.66 | 61.09 | 57.58 | 52.46 |
| | QuIP# | 3 | 59.51 | 34.22 | 59.23 | 61.09 | 65.20 | 55.85 |
| | OmniQuant | 3 | 35.69 | 25.77 | 35.48 | 52.88 | 42.48 | 38.46 |
| Llama 3 70B | Baseline | 16 | 85.23 | 64.33 | 84.07 | 77.66 | 86.27 | 79.51 |
| | **S-Quant** | **3.8** | **83.94** | **59.31** | **83.89** | **77.80** | **85.62** | **78.11** |
| | SeedLM | 4 | 83.80 | 59.30 | 83.84 | 77.74 | 85.60 | 78.06 |
| | AWQ | 4 | 80.98 | 57.94 | 82.84 | 60.54 | 79.39 | 72.34 |
| | QuIP# | 4 | OOM | OOM | OOM | OOM | OOM | OOM |
| | OmniQuant | 4 | 25.13 | 26.54 | 26.36 | 51.38 | 37.83 | 33.45 |
| | **S-Quant** | **2.7** | **78.50** | **52.24** | **80.83** | **77.48** | **84.66** | **74.74** |
| | SeedLM | 3 | 78.45 | 52.22 | 80.77 | 77.35 | 84.59 | 74.68 |
| | AWQ | 3 | 65.87 | 45.14 | 70.76 | 55.88 | 69.08 | 61.35 |
| | QuIP# | 3 | OOM | OOM | OOM | OOM | OOM | OOM |
| | OmniQuant | 3 | 25.21 | 25.94 | 26.15 | 49.64 | 37.83 | 32.95 |
| Qwen2.5 7B | Baseline | 16 | 79.56 | 46.08 | 78.93 | 72.38 | 79.35 | 71.26 |
| | **S-Quant** | **3.8** | **74.13** | **45.17** | **75.47** | **69.33** | **77.83** | **68.39** |
| | SeedLM | 4 | 74.07 | 45.03 | 75.29 | 69.11 | 77.67 | 68.23 |
| | AWQ | 4 | 71.09 | 44.31 | 75.13 | 69.01 | 78.63 | 67.63 |
| | QuIP# | 4 | 68.73 | 40.27 | 72.93 | 66.01 | 75.59 | 64.71 |
| | OmniQuant | 4 | 71.13 | 43.83 | 74.63 | 68.61 | 74.07 | 66.45 |
| | **S-Quant** | **2.7** | **70.43** | **41.87** | **71.27** | **66.93** | **74.83** | **65.07** |
| | SeedLM | 3 | 70.31 | 41.73 | 71.03 | 66.71 | 74.61 | 64.88 |
| | AWQ | 3 | 53.83 | 34.03 | 57.07 | 61.63 | 58.03 | 52.92 |
| | QuIP# | 3 | 59.83 | 34.63 | 59.73 | 61.63 | 65.43 | 56.25 |
| | OmniQuant | 3 | 36.03 | 26.03 | 35.73 | 53.13 | 42.73 | 38.73 |

*Table 1.* Performance comparison across different models and zero-shot tasks for around 4-bit and 3-bit configurations. Entries that ran out of memory in our setup are marked with OOM.

| Method | Bits | 2-7B | 2-13B | 2-70B | 3-8B | 3-70B |
|---|---|---|---|---|---|---|
| Baseline | 16 | 5.5 | 4.9 | 3.3 | 6.1 | 2.9 |
| **S-Quant** | **3.8** | **5.7** | **5.0** | **3.5** | **6.8** | **3.6** |
| SeedLM | 4 | 5.7 | 5.1 | 3.5 | 7.0 | 3.8 |
| OmniQuant | 4 | 6.1 | 5.2 | 3.7 | inf | inf |
| AWQ | 4 | 5.8 | 5.1 | 3.5 | 7.1 | 4.7 |
| QuIP# | 4 | 6.5 | 5.3 | OOM | 7.6 | OOM |
| **S-Quant** | **2.7** | **6.5** | **5.8** | **3.8** | **9.7** | **5.4** |
| SeedLM | 3 | 6.6 | 5.8 | 4.0 | 10.1 | 5.7 |
| OmniQuant | 3 | inf | 10.7 | 7.5 | inf | inf |
| AWQ | 3 | 15.6 | 6.5 | 4.4 | 11.8 | 11.6 |
| QuIP# | 3 | 10.8 | 5.7 | OOM | 10.1 | OOM |

*Table 2.* WikiText-2 perplexities for LLaMA 2, LLaMA 3 using 3- and 4-bit weight representations evaluated on 2048-token contexts.

| MODULE | NUMBER | AREA (mm$^2$) | RATIO (%) |
|---|---|---|---|
| PE (121.53 $\mu$m$^2$) | 256 | 0.0311 | 97.2 |
| GENERATOR (912.23 $\mu$m$^2$) | 1 | 0.0009 | 2.8 |

*Table 3.* Area breakdown.

comprises several multipliers and a cache-backed look-up table (LUT) that stores all linear-feedback shift register (LFSR) states. Next, we evaluated the area of the systolic array: each processing element (PE) contains a multiplier, an adder, and a local buffer. As shown in Table 3, for a $16 \times 16$ systolic array accelerator, the weight generator occupies only a small fraction of the overall area—approximately 3%. This indicates that the area overhead introduced by the generator in S-Quant is negligible.

**Latency and Energy**. During latency and energy measurements, we compared S-Quant (the average 3.8-bit case) against three baselines: an uncompressed weight design,

Llama3 model families ranging from 7B to 70B parameters. We present a systematic comparison of the four methods with respect to latency, energy efficiency, and silicon area.

**Area**. We analyzed the area of the weight generator, which

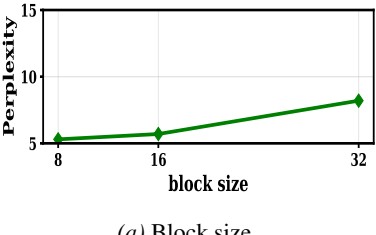 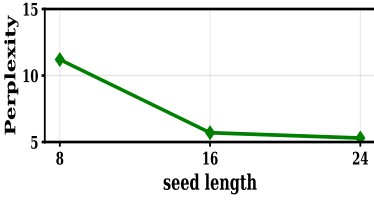 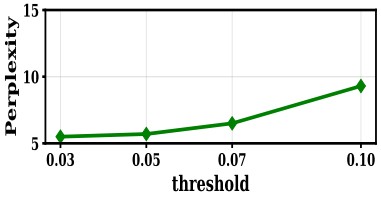

*(a)* Block size.  *(b)* Seed length.  *(c)* Explained energy ratio threshold.

*Figure 5.* Ablation on LLaMA2-7B: effect of block size, seed length, and explained energy ratio threshold on perplexity (PPL), varying one factor at a time while fixing the others.

seed-compression (Shafipour et al., 2024) and a 4-bit weight-quantization design. To ensure an area-fair comparison among the three configurations, we adjusted the number of PEs in all four cases so that they have the same total area. As shown in Figure 6a, relative to the uncompressed baseline S-Quant achieves an approximately $4\times$ reduction in latency across multiple benchmarks and also outperforms the 4-bit quantized design in execution time. Figure 6b reports energy consumption: weight compression substantially reduces memory-access power, producing large savings in DRAM and SRAM energy; the figure also presents a breakdown of power into static, DRAM, SRAM, and core components. In addition, since seed compression involves a large number of floating-point multipliers, it exhibits higher latency and power consumption compared to S-Quant.

### 4.3. Ablation Study

**Block size, Seed length and Explained energy ratio.** Figure 5a, Figure 5b, and Figure 5c present an ablation study that analyzes the impact of three key hyperparameters—*threshold*, *seed length*, and *block size*—on model performance measured by perplexity (PPL) on LLaMA2-7B. In this study, we adopt a controlled experimental protocol in which only one variable is changed at a time, while all other settings are kept fixed, allowing us to isolate the effect of each factor. The results clearly show that all three variables have a significant influence on perplexity. Specifically, varying the threshold leads to noticeable changes in PPL, indicating that the choice of energy or reconstruction threshold directly affects model quality. Similarly, seed length plays a critical role, as different initialization lengths result in substantial performance variation. Block size also exhibits a strong impact, suggesting that the granularity of block partitioning is closely tied to the effectiveness of the method. These observations demonstrate that threshold, seed length, and block size are not minor implementation details, but rather fundamental design choices that can dramatically affect outcomes. Consequently, these variables must be explicitly incorporated into the design space exploration and carefully tuned to achieve optimal performance.

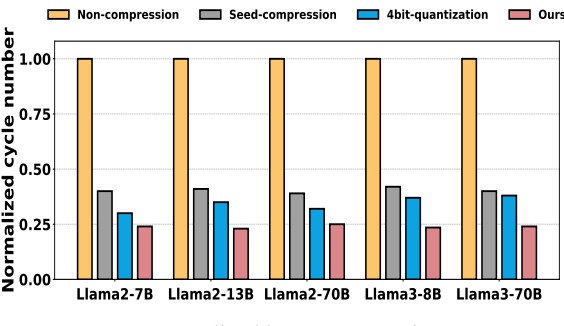

*(a)* Normalized latency comparison.

*(b)* Normalized energy breakdown.

*Figure 6.* Performance comparison of compression techniques: (a) Normalized execution cycles. (b) Normalized energy consumption. Both normalized to non-compression baseline.

## 5. Conclusion

We present S-Quant, a novel weight-only compression method that reconstructs each weight block from a compact LFSR seed and a small set of basis tensors, substantially reducing stored weights and chip-to-chip bandwidth. We adapt the number of basis tensors per block with an explained-energy metric to trade off reconstruction error and compression, and we treat block size, seed length, and related hyperparameters as a multi-objective design-space exploration solved via Bayesian optimization. We implement a hardware accelerator in SystemVerilog and demonstrate in simulation up to a 4× speedup for memory-bound LLM inference. Extensive experiments on LLM models (7B–70B) show state-of-the-art accuracy at roughly 3–4

bit effective precision. S-Quant paves the way for new hardware-friendly compression techniques for LLM.

## Impact Statement

This paper presents work whose goal is to advance the field of Machine Learning. There are many potential societal consequences of our work, none which we feel must be specifically highlighted here.

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

# A. Experiments

| Model | Method | Bits | ARC-Easy | ARC-Challenge | HellaSwag | WinoGrande | BoolQ | Mean |
|-------|--------|------|----------|---------------|-----------|------------|-------|------|
| Llama 2 13B | Baseline | 16 | 77.44 | 48.98 | 79.38 | 72.22 | 80.55 | 71.71 |
| | **S-Quant(Ours)** | 3.8 | 77.02 | 49.93 | 78.55 | 72.81 | 79.33 | 71.53 |
| | SeedLM | 4 | 76.98 | 49.83 | 78.54 | 72.77 | 79.20 | 71.46 |
| | AWQ | 4 | 77.44 | 49.32 | 78.57 | 71.90 | 78.47 | 71.14 |
| | QuIP# | 4 | 74.24 | 45.48 | 77.17 | 71.27 | 79.51 | 69.53 |
| | OmniQuant | 4 | 76.18 | 47.95 | 78.10 | 72.14 | 81.77 | 71.23 |
| | **S-Quant(Ours)** | 2.7 | 72.96 | 45.43 | 74.62 | 71.51 | 78.81 | 68.67 |
| | SeedLM | 3 | 72.85 | 45.39 | 74.50 | 71.35 | 78.81 | 68.58 |
| | AWQ | 3 | 70.58 | 45.14 | 72.72 | 64.96 | 72.45 | 65.17 |
| | QuIP# | 3 | 73.48 | 45.14 | 74.92 | 69.06 | 79.60 | 68.44 |
| | OmniQuant | 3 | 55.85 | 34.47 | 59.54 | 53.04 | 63.39 | 53.26 |
| Llama 2 70B | Baseline | 16 | 80.98 | 57.25 | 83.81 | 77.98 | 83.70 | 76.74 |
| | **S-Quant(Ours)** | 3.8 | 81.30 | 56.54 | 83.04 | 76.75 | 82.45 | 76.02 |
| | SeedLM | 4 | 81.14 | 56.40 | 82.97 | 76.72 | 82.26 | 75.90 |
| | AWQ | 4 | 80.98 | 56.66 | 83.24 | 77.19 | 83.27 | 76.27 |
| | QuIP# | 4 | OOM | OOM | OOM | OOM | OOM | OOM |
| | OmniQuant | 4 | 79.59 | 55.97 | 82.67 | 76.80 | 83.43 | 75.69 |
| | **S-Quant(Ours)** | 2.7 | 79.07 | 53.86 | 80.53 | 76.97 | 79.14 | 73.91 |
| | SeedLM | 3 | 79.00 | 53.84 | 80.51 | 76.80 | 79.02 | 73.83 |
| | AWQ | 3 | 80.26 | 55.80 | 80.50 | 73.01 | 80.00 | 73.91 |
| | QuIP# | 3 | OOM | OOM | OOM | OOM | OOM | OOM |
| | OmniQuant | 3 | 63.59 | 39.51 | 68.24 | 62.04 | 65.23 | 59.72 |
| Llama 3 8B | Baseline | 16 | 76.81 | 52.73 | 76.97 | 72.93 | 81.87 | 72.26 |
| | **S-Quant(Ours)** | 3.8 | 76.68 | 49.89 | 76.72 | 73.12 | 80.84 | 71.45 |
| | SeedLM | 4 | 76.52 | 49.74 | 76.61 | 72.93 | 80.76 | 71.31 |
| | AWQ | 4 | 74.49 | 51.54 | 78.03 | 73.09 | 80.40 | 71.51 |
| | QuIP# | 4 | 72.39 | 46.93 | 75.93 | 71.82 | 79.24 | 69.26 |
| | OmniQuant | 4 | 73.95 | 47.78 | 73.42 | 69.69 | 71.99 | 67.37 |
| | **S-Quant(Ours)** | 2.7 | 67.32 | 41.72 | 68.46 | 69.39 | 67.73 | 62.92 |
| | SeedLM | 3 | 67.21 | 41.55 | 68.34 | 69.22 | 67.61 | 62.79 |
| | AWQ | 3 | 64.90 | 40.19 | 68.40 | 65.04 | 74.62 | 62.63 |
| | QuIP# | 3 | 65.07 | 40.36 | 67.79 | 68.82 | 72.14 | 62.84 |
| | OmniQuant | 3 | 30.26 | 22.53 | 28.96 | 49.33 | 48.47 | 35.91 |

*Table 4.* Performance comparison across different models and zero-shot tasks for around 4-bit and 3-bit configurations. Entries that ran out of memory in our setup are marked with OOM.

# B. S-Quant Reconstruction Overhead

Figure 7 illustrates the end-to-end dataflow of the proposed accelerator. During execution, the compact seeds and coefficients are first fetched from memory and fed into the weight reconstruction module. Leveraging the seed as an index, the generator performs a single-cycle lookup into a precomputed table to retrieve the corresponding tensor representation, followed by on-the-fly weight reconstruction using parallel multiplier and adder units. The reconstructed weights are then streamed directly into the systolic array to support efficient multiply–accumulate (MAC) operations without intermediate buffering.

To quantify the runtime cost of weight reconstruction, we incorporate the reconstruction logic into our cycle-accurate simulator and evaluate its contribution to the total execution time under different batch sizes. Our results show that the relative reconstruction overhead decreases as the batch size increases, and becomes negligible at larger batch sizes. This observation indicates that the reconstruction process introduced by S-Quant incurs an acceptable runtime cost and does not pose a performance bottleneck in practical deployments.

# C. Ablation Study

**Importance of adaptive basis selection.** As illustrated in Figure 8 9, we compare fixed and adaptive base allocation strategies under two different fixed settings. In the first experiment, each block is assigned a fixed number of two bases, while in the second experiment the fixed number is increased to four. For the fixed strategy, all blocks share the same number

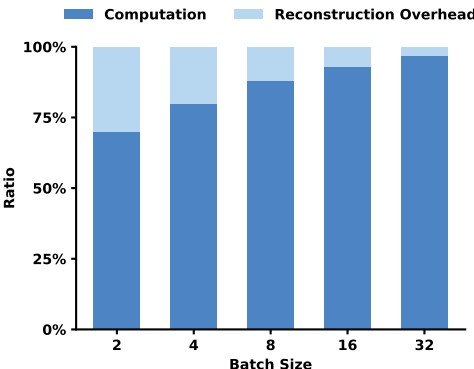

*Figure 7.* S-Quant Reconstruction Overhead.

of bases regardless of their intrinsic complexity. As a result, the explained energy ratios vary significantly across blocks: some blocks are over-allocated and achieve energy ratios far above the target threshold $R_{\text{th}}$, whereas others fail to reach the threshold, leading to insufficient reconstruction quality. Increasing the fixed number of bases alleviates this issue to some extent, but cannot fundamentally guarantee that all blocks satisfy the target requirement. In contrast, the adaptive strategy dynamically selects the number of bases for each block according to its energy distribution. As shown in Figure 8 9, the adaptive approach consistently ensures that the explained energy ratio of every block exceeds the target threshold, resulting in more balanced reconstruction quality and more efficient utilization of bases across blocks.

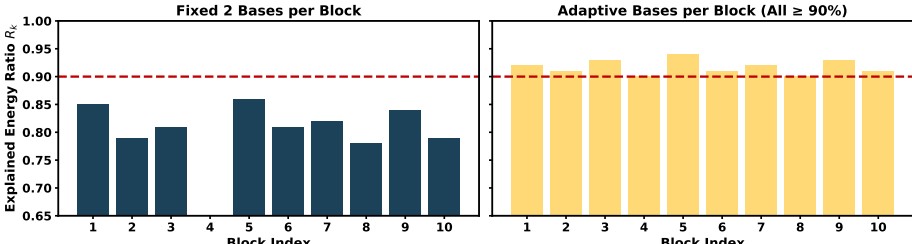

*Figure 8.* Fixed 2 based vs. Adaptive.

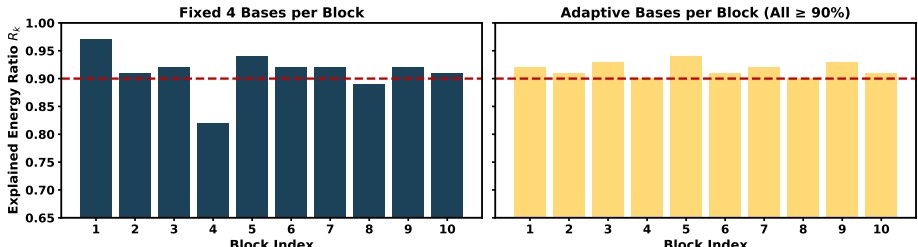

*Figure 9.* Fixed 4 based vs. Adaptive.

We provide the details of the multi-objective Bayesian optimization (MOBO) framework employed to efficiently explore the design space and approximate the Pareto frontier.

**Problem formulation.** Each design configuration $\mathbf{x} \in \mathcal{D}$ is associated with a vector-valued objective function

$$\mathbf{f}(\mathbf{x}) = \big(f_1(\mathbf{x}), f_2(\mathbf{x})\big), \tag{17}$$

where $f_1(\mathbf{x})$ denotes reconstruction error and $f_2(\mathbf{x})$ denotes storage cost. These objectives are inherently conflicting, and thus the goal of optimization is to identify a set of Pareto-optimal configurations

$$\mathcal{P}^\star = \{\mathbf{x} \in \mathcal{D} \mid \nexists \, \mathbf{x}' \in \mathcal{D} : \mathbf{f}(\mathbf{x}') \prec \mathbf{f}(\mathbf{x})\}, \tag{18}$$

where $\prec$ denotes Pareto dominance.

**Gaussian process surrogate models.** We model each objective independently using a Gaussian Process (GP),

$$f_m(\mathbf{x}) \sim \mathcal{GP}\big(m_m(\mathbf{x}), k_m(\mathbf{x}, \mathbf{x}')\big), \quad m \in \{1, 2\}, \tag{19}$$

Given a set of observed configurations, the posterior predictive distribution for each objective is

$$p\big(f_m(\mathbf{x}) \mid \mathcal{D}_N\big) = \mathcal{N}\big(\mu_m(\mathbf{x}), \nu_m(\mathbf{x})\big), \tag{20}$$

This independent modeling assumption is commonly adopted in MOBO and provides a favorable trade-off between modeling expressiveness and computational efficiency.

**Hypervolume indicator.** To evaluate the quality of a Pareto set, we adopt the hypervolume (HV) indicator. Given a reference point $\mathbf{r}$, the hypervolume of a Pareto set $\mathcal{P}$ is defined as

$$HV(\mathcal{P}) = \lambda\left(\bigcup_{\mathbf{x} \in \mathcal{P}} [\mathbf{f}(\mathbf{x}), \mathbf{r}]\right), \tag{21}$$

where $\lambda(\cdot)$ denotes the Lebesgue measure. Hypervolume is a strictly Pareto-compliant metric that jointly captures convergence and diversity of the Pareto frontier.

**Expected hypervolume improvement.** Let $\mathcal{P}_N$ denote the current set of non-dominated solutions after $N$ evaluations. For a candidate configuration $\mathbf{x}$, the hypervolume improvement is defined as

$$\Delta HV(\mathbf{x}) = HV\big(\mathcal{P}_N \cup \{\mathbf{f}(\mathbf{x})\}\big) - HV(\mathcal{P}_N), \tag{22}$$

Since $\mathbf{f}(\mathbf{x})$ is unknown prior to evaluation, the Expected Hypervolume Improvement (EIPV) acquisition function is defined as

$$EIPV(\mathbf{x}) = \mathbb{E}_{\mathbf{f}(\mathbf{x})}\big[\Delta HV(\mathbf{x})\big], \tag{23}$$

where the expectation is taken with respect to the GP posterior distributions of the objectives.

**Bi-objective decomposition.** In the bi-objective setting considered in this work, EIPV admits an efficient decomposition over non-overlapping hyper-rectangular regions induced by the current Pareto frontier and the reference point. Assuming independent GP posteriors for $f_1$ and $f_2$, the expectation factorizes as

$$EIPV(\mathbf{x}) = \sum_k \int_{\mathcal{R}_k} p\big(f_1(\mathbf{x})\big) p\big(f_2(\mathbf{x})\big) \, df_1 \, df_2, \tag{24}$$

where $\{\mathcal{R}_k\}$ denotes the set of dominated regions in the objective space. Each term corresponds to the expected contribution of $\mathbf{x}$ to the hypervolume expansion.

**Sampling strategy.** At each iteration, the next design configuration is selected by maximizing the EIPV acquisition function as

$$\mathbf{x}^\star = \arg\max_{\mathbf{x} \in \mathcal{D}} EIPV(\mathbf{x}), \tag{25}$$

which explicitly favors configurations that are expected to expand the current Pareto frontier under a limited evaluation budget.

