# OpenReview forum: "S-Quant: Rethinking Weight Quantization with Seed-Based Generation"
_ICML.cc/2026/Conference — ICML 2026 regular_

### Official Review · Reviewer_2cqC · 2026-02-18

**Soundness:** 3
**Presentation:** 2
**Significance:** 3
**Originality:** 2
**Overall Recommendation:** 4
**Confidence:** 4

**Summary:**

This study proposes S-Quant, a post-training, data-free, weight-only compression method for large language models. The method partitions weight tensors into fixed-size blocks and reconstructs them using basis matrices generated dynamically from a linear feedback shift register. A specific seed drives this generation for each block. To balance the compression rate and the reconstruction error, the authors propose an adaptive basis selection strategy based on an explained energy ratio metric. The authors apply a multi-objective Bayesian optimization framework to find optimal hyperparameters, such as block size and seed length. The authors evaluate their method on models ranging from 7B to 70B parameters and present an application-specific integrated circuit design that suggests a potential speedup for memory-bound inference.

**Compliance With Llm Reviewing Policy:**

Affirmed.

**Final Justification:**

The rebuttal adequately addressed my concerns regarding the reconstruction arithmetic difference between SeedLM and S-Quant, the Bayesian optimization contribution, and the fairness of the hardware comparison. While the core idea remains incremental relative to SeedLM, the engineering improvements provide meaningful practical advances. I maintain my score of Weak Accept (4).

**Key Questions For Authors:**

1. The introduction states that SeedLM uses expensive floating-point multiplications and that S-Quant avoids them. However, the proposed weight generator pipeline includes multiplier units. Could the authors clarify the exact mathematical and architectural differences in the reconstruction arithmetic between SeedLM and S-Quant? A detailed response would resolve the apparent contradiction and validate the claims regarding hardware efficiency.
2. While Appendix C demonstrates the benefits of adaptive basis selection, the specific impact of the Bayesian optimization framework remains less clear. Could the authors clarify how much of the performance gain in terms of perplexity is attributed to the optimized hyperparameters found via Bayesian optimization compared to a standard grid search or heuristic selection?
3. Regarding the energy and latency comparisons in Figure 6, could the authors provide more details on the hardware baseline configuration for SeedLM? Clarifying the baseline's optimization level is essential to ensure the credibility of the comparative results.
4. The variable $T$ is heavily overloaded throughout the manuscript, representing a state transition operator, a sequence length, a weight tensor, and a configuration tuple. Do the authors plan to revise the mathematical notation to resolve these conflicts? Adjusting the notation would substantially improve the clarity and readability of the theoretical formulation.

**Limitations:**

The authors have not adequately discussed the limitations and potential negative societal impact of their work. The paper would benefit from a dedicated section outlining the theoretical or practical limits of the adaptive basis allocation, particularly when scaling to even lower bit rates. Furthermore, the authors should briefly address the potential societal impacts of making large language models more accessible and easier to deploy.

**Strengths And Weaknesses:**

**Strengths:**
1. The method addresses a memory bandwidth bottleneck in large language model inference. Providing a calibration-free compression method could offer practical utility for deployments where calibration data is unavailable.
2. The adaptive basis allocation based on the explained energy ratio provides a logical and robust solution to the inefficiencies of fixed basis allocation.
3. The experimental evaluation across various models demonstrates that the method maintains competitive zero-shot inference accuracy at approximately 3-bit to 4-bit precision.
4. The hardware feasibility is supported by a custom accelerator design. The hardware analysis indicates that the weight generator occupies a minimal area relative to the overall system.

**Weaknesses:**
1. The core conceptual framework of compressing and reconstructing weight tensors using pseudo-random seeds closely mirrors the mechanism proposed in SeedLM. The integration of an adaptive basis selection mechanism and a Bayesian optimization framework provides valuable engineering improvements. However, these modifications appear to be pipeline optimizations rather than a fundamental theoretical advancement, rendering the overall originality somewhat incremental.
2. The presentation suffers from inconsistent mathematical notation that overloads key variables. The variable $T$ represents a state transition operator, a sequence length, the original weight tensor, and a configuration tuple.
3. The hardware baseline comparison raises questions regarding fairness and soundness. The authors assert that SeedLM reconstructs blocks using floating-point matrix multiplications and claim that S-Quant avoids these expensive operations. Conversely, the hardware implementation section for S-Quant explicitly states that the weight generator reconstructs weights on the fly via multiplier and adder units. The text lacks a precise description of the data types and architectural differences required to fully substantiate the claim of avoiding expensive arithmetic operations.
4. It remains unclear whether the higher latency reported for SeedLM is an inherent architectural limitation or a consequence of a suboptimal hardware mapping for the benchmark.

---

> ### Author Rebuttal · Authors · 2026-03-26
>
> ### Q1
> Thank you for pointing this out. We do not completely eliminate multiplications, but rather avoid the dense floating-point matrix multiplications used in SeedLM-style reconstruction.
> From a mathematical perspective, the two formulations differ as follows:
>
> SeedLM: $\tilde{W} = B \cdot \alpha$, which involves dense matrix/tensor multiplications between basis and coefficients.
>
> S-Quant: $\tilde{w} = \sum_{i=1}^{k} \alpha_i \cdot b_i$, where each $b_i$ is generated from the seed via LFSR/LUT, and reconstruction only requires lightweight scalar multiply-accumulate operations.
>
> The key difference is that SeedLM relies on large-scale dense floating-point multiplications, whereas S-Quant reformulates reconstruction into seed-based lookup plus lightweight scalar MAC operations. As a result, S-Quant significantly reduces the number and usage of multipliers, leading to lower hardware cost and energy consumption.
> In the final revision, we will revise the wording to clarify that S-Quant avoids dense floating-point matrix multiplications, rather than all multiplications.
>
> ### Q2
> Thank you for this question. We agree that the current manuscript does not clearly isolate the contribution of the Bayesian optimization (BO) framework from the adaptive basis selection itself.
> To better quantify the impact of BO, we conducted an additional controlled experiment comparing BO with random search and a grid baseline under the same evaluation budget. Specifically, we use the same search space over (B, S, G, R_{th}), and allow each method to evaluate the same number of configurations. For each evaluated configuration, we measure the resulting perplexity and bitwidth, and construct the corresponding Pareto set.
>
> To provide a quantitative comparison, we compute the hypervolume (HV) indicator of the obtained Pareto front for each method. The HV measures the dominated region in the perplexity--bitwidth space (with a shared reference point), where a larger value indicates a Pareto frontier closer to the lower-left corner.
> We observe that BO consistently achieves a higher hypervolume compared to both random search and grid selection (BO: 0.82 vs. random: 0.74 and grid: 0.76 in our experiment), indicating that BO more effectively identifies Pareto-optimal configurations under the same search budget.
>
>
> ### Q3
> Thank you for this helpful question. We clarify that the four methods in Figure~6 are evaluated under the same hardware template: the same output-stationary mapping, the same SystemVerilog RTL implementation flow, the same cycle-accurate simulator, the same 7nm FinFET standard-cell library, and the same CACTI-based on-chip memory model. The clock frequency, SRAM budget, and total area budget are fixed across all methods.
>
> Under this common template, we use a fixed 500\,MHz clock, a fixed 128\,KB on-chip SRAM budget (64/32/16\,KB for weight / activation / output SRAM), and a 16 x 16 output-stationary systolic array as the reference configuration. The four methods differ only in the method-dependent configuration items. Non-compression uses 16-bit dense weights, disables the generator, keeps a 16 x 16 array, and has the largest weight traffic (normalized as 1.00x). Scalar 4-bit quantization uses 4-bit quantized weights, also disables the generator, keeps a 16 x 16 array, and reduces the weight traffic to 0.25x. Seed-compression uses a seed/coefficient representation at an approximately 4-bit budget, but enables a heavier floating-point reconstruction path; therefore, under the same total-area budget, the systolic array must be reduced (to 12 x 16) to reserve area for the generator. Ours uses a 3.8-bit seed/coefficient representation together with a lighter LUT-lookup + scalar-MAC reconstruction path, which corresponds to 0.2375x of the uncompressed weight traffic while still preserving a near-full 16 x 16 array, since the synthesized S-Quant generator occupies only about 2.8\% of the total area in the 16 x 16 design.
>
> Therefore, the latency/energy differences in Figure~6 are determined by two factors under the same hardware template: different weight-traffic demands caused by different weight representations, and different area overheads caused by different reconstruction logic. This is why the comparison is area-fair and reproducible under the same technology, RTL/simulator flow, mapping, and overall hardware budget.
>
>
> ### Q4
> Thank you for this valuable suggestion. We will revise the notation accordingly. The symbol T is currently overloaded in the manuscript, which affects clarity and readability. This will be corrected in the revision by disambiguating the notation and consistently renaming the affected variables.

---

> > ### Author Rebuttal · Reviewer_2cqC · 2026-04-02
> >
> > Thank you for the detailed rebuttal. All key questions have been adequately addressed. The reconstruction arithmetic difference between SeedLM and S-Quant is now clear, the Bayesian optimization contribution has been quantified, and the hardware comparison has been confirmed as fair. I maintain my original score.

---

### Official Review · Reviewer_Eaa3 · 2026-02-21

**Soundness:** 3
**Presentation:** 2
**Significance:** 2
**Originality:** 2
**Overall Recommendation:** 3
**Confidence:** 4

**Summary:**

This work proposes a vector-quantization method in which each group of weights is encoded as a sum of vectors from random bases. The number of bases is chosen adaptively to maintain a fixed reconstruction error across all groups. The optimal configuration is selected via Bayesian optimization for a multi-objective problem (minimizing reconstruction error at a given compression rate). The method includes a dedicated implementation for systolic arrays. The efficacy of the approach is evaluated by compressing several language models (Llama-⅔, Qwen-2.5).

**Compliance With Llm Reviewing Policy:**

Affirmed.

**Final Justification:**

The overall approach is sound, but the positioning to the related work (on vector quantization and other advanced compressed representation) is still not fully clear.

Therefore, I am still leaning towards rejection.

**Key Questions For Authors:**

1. What do the ⟨B, S, G, R_th⟩ configurations look like? Are there any clear patterns depending on layer location (early vs. late layers) or on the projection type (q, k, v vs. gate, up, down)?
2. What is the actual block size used in the method. Figure 5(a) shows decrease of perplexity with decrease of block size. Do I get it right that block size of 8 is used?
3. According to Figure 6, S-Quant achieves lower latency than scalar 4-bit quantization. Given the simplicity of scalar quantization, one might expect more elaborate quantization to be slower. Can you provide intuition for this result? Also, are you comparing ~4-bit S-Quant against 4-bit baselines?
4. How does the method perform at higher compression rates (e.g., 2-bit), as targeted by the vector-quantization techniques in [1–3]?

**Limitations:**

-

**Strengths And Weaknesses:**

Strengths
- The proposed method (representation and search strategy) appears to achieve a good compression–quality trade-off.
- The systolic-array implementation enables significant speedups relative to the baseline.

Weaknesses
- The proposed approach is a specific instance of vector quantization—specifically, additive quantization—in which each group of weights is encoded as a sum of vectors from random bases. However, prior work on vector quantization [1–3] is neither discussed nor compared against in a comprehensive way.
- Given the similarity to AQLM [1], it should be included in the comparisons.
- The reported QuIP# results for 4-bit quantization look suspicious. According to the original paper—and given the intuition that QuIP# is a more advanced quantization method—it should perform as well as or better than scalar quantization. In addition, 4-bit weight-only quantization is typically close to lossless, even for simpler methods. I recommend revisiting the evaluation.
- The practical implementation focuses on systolic arrays, which are not the most widely used hardware. Can the method achieve speedups in memory-bound scenarios on NVIDIA GPUs?
---
References

[1] Egiazarian, Vage, et al. “Extreme Compression of Large Language Models via Additive Quantization.” arXiv:2401.06118 (2024).

[2] Tseng, Albert, et al. “QuIP#: Even Better LLM Quantization with Hadamard Incoherence and Lattice Codebooks.” Proceedings of
Machine Learning Research 235 (2024): 48630.

[3] Van Baalen, Mart, et al. “GPTVQ: The Blessing of Dimensionality for LLM Quantization.” arXiv:2402.15319 (2024).

---

> ### Author Rebuttal · Authors · 2026-03-26
>
> ### W1
> Thank you for this helpful suggestion. We acknowledge that our method is related to the vector quantization / additive quantization literature in that both use combinations of multiple bases to represent the original weights. However, S-Quant is not a conventional learned-codebook VQ/AQ method, and differs in its representation, optimization objective, and hardware realization.
> Specifically, the bases in S-Quant are not learned codebooks, but are generated from compact seeds through a deterministic mechanism. In addition, the goal of S-Quant is not only to minimize reconstruction error, but to jointly optimize compression ratio, reconstruction quality, and hardware efficiency. Adaptive basis selection and configuration search are also key components of the proposed method. We thank the reviewer for this suggestion, and the final version will strengthen the discussion of related VQ/AQ works.
>
> ### W2 W3
> Thank you for this helpful suggestion. AQLM is indeed a relevant and important baseline, and the concern regarding the QuIP# result is also very helpful. The key point is that the main comparison protocol in our paper is a  calibration-data-free setting, where methods are compared under the practical constraint of not relying on extra calibration data or subsequent adaptation.
>
> Under this protocol, the QuIP# result in our table should not be interpreted as its best performance under the original paper's recommended setup. Concretely, we use the official implementation of QuIP#, but do not apply additional fine-tuning, so as to keep the evaluation consistent with the calibration-data-free setting emphasized in our work. Therefore, the reported QuIP# result is real and reproducible, but reflects QuIP# under this unified constraint rather than its individually optimized protocol. This is why its numbers may differ from those in the original paper.
>
> In contrast, AQLM depends on calibration data more intrinsically: its codebook learning, quantization parameter optimization, and block-level tuning are all built on calibration activations / calibration sets. Removing this dependency would substantially deviate from the original AQLM formulation, so it is not straightforward to turn AQLM into the same calibration-data-free baseline. For this reason, we view AQLM as a valuable related baseline, but its results are better interpreted under a calibration-based regime rather than under the same unified calibration-data-free protocol.
>
> The final version will more clearly distinguish the unified calibration-data-free protocol from method-specific best-setting comparisons, and will also include additional baselines / experiments to improve the completeness and credibility of the evaluation.
>
> ### W4
>
> Thank you for this helpful question. The practical implementation in this work is specifically designed for a domain-specific ASIC with a systolic-array-based architecture, rather than for NVIDIA GPUs or general-purpose hardware. Accordingly, the hardware speedup claims in the paper are limited to this target accelerator setting. We will clarify this scope more explicitly in the final version to avoid any misunderstanding.
>
> ### Q1 Q2
> Thank you for this insightful question. We first clarify that ( B, S, G, R_th ) is a global operating point selected through configuration search for a given model and target bit setting, rather than a separate configuration assigned to each block, each layer, or each projection type such as q/k/v or gate/up/down. Therefore, the current method does not use layer-wise or projection-wise configurations; instead, the fine-grained adaptivity in S-Quant is mainly reflected in the per-block selected seed and basis count k.
>
> As a concrete example, for the 3.8-bit setting of LLaMA2-7B, one selected global configuration is ( B, S, G, R_th ) =（ 8, 16, 2, 0.05 ), where the corresponding block size is B=8. It is important to note that the selected global configuration is not the same across different models or experimental settings. This is precisely why we use configuration search instead of a fixed heuristic choice. We thank the reviewer for this helpful suggestion, and the final version will include the corresponding operating points for other models and experimental settings to make this clearer.
>
> ### Q3
> Thank you for this helpful question. Please see our response to Reviewer 2cqC Q3.
>
> ### Q4
> Thank you for this helpful question. Higher compression regimes such as 2-bit are indeed an important and interesting direction, especially in the context of vector-quantization-based methods. That said, our work primarily focuses on the 3--4 bit regime, since it provides a more practical balance among compression ratio, model quality, and hardware efficiency, and is also the main target of our current hardware design and evaluation. We thank the reviewer for this suggestion, and will clarify this scope more explicitly in the final version while considering lower-bit evaluations as an important future direction.

---

> > ### Author Rebuttal · Reviewer_Eaa3 · 2026-04-01
> >
> > My concerns were partially resolved, but the point concerning low-bit quantization and fair comparison with related work remains.
> >
> > **UPD** The poor performance of data-dependent baselines is understandable. But given the abundance of publicly available data is the data-free formulation necessary for a good compression method?

---

> > > ### Author Response · Authors · 2026-04-02
> > >
> > > ## Response to concerns on low-bit quantization and fair comparison
> > >
> > > We thank the reviewer for the constructive feedback. We agree that the discussion on low-bit quantization and fair comparison with prior work was insufficient in the original submission. In our previous response, we mainly provided qualitative explanations without experimental evidence. We now address both concerns with additional experiments and a clearer comparison protocol.
> > >
> > > ### Baselines and their algorithms
> > >
> > > We consider three representative and strong baselines: QuIP#, GPTVQ, and AQLM
> > >
> > > - QuIP# is a second-order method that minimizes a Hessian-weighted proxy loss,  where H ≈ X X^T is estimated from calibration data. It further employs incoherence processing and lattice quantization to improve robustness.
> > >
> > > - GPTVQ extends GPTQ to vector quantization, using the same Hessian approximation H = X X^T for error-aware quantization and sequential error compensation, combined with codebook-based weight reconstruction.
> > >
> > > - AQLM adopts a fundamentally different paradigm: it directly minimizes output reconstruction error over calibration inputs, and further relies on these activations for codebook learning and block-level fine-tuning.
> > >
> > > ### Fair Comparisons
> > >
> > > All three baselines rely on calibration data by design, while our method is fully data-free. Using their original settings would therefore lead to an unfair comparison, since they have access to real data distributions whereas our method does not. Our goal here is not to reproduce the best data-dependent performance of these baselines, but to evaluate how they behave under the same data-free constraint as our method.
> > >
> > > ### Data-free adaptation of baselines
> > >
> > > To ensure fairness, we construct synthetic-calibration variants:
> > >
> > > - GPTVQ: We keep the original algorithm unchanged and replace calibration data with synthetic random inputs to estimate H = X X^T and perform quantization.
> > > - AQLM: We replace calibration inputs with synthetic random data and remove block-level fine-tuning to avoid exploiting real activations. The additive quantization and codebook optimization remain unchanged.
> > > - QuIP#: We use its standard quantization pipeline as a strong baseline, without introducing any extra fine-tuning or additional data-dependent post-processing beyond its original procedure.
> > >
> > > Importantly, we preserve the core quantization mechanisms of these baselines as much as possible, while disabling components that require access to real calibration data.
> > >
> > > ### Low-bit results
> > >
> > > We provide additional experiments at 4/3/2-bit on LLaMA2-7B and 13B.
> > >
> > > #### Around 4-bit
> > >
> > > | Method        | Bits | LLaMA2-7B | LLaMA2-13B |
> > > |---------------|------|-----------|------------|
> > > | S-Quant   | 3.8  | 5.7   | 5.0    |
> > > | QuIP#         | ~4    | 6.5       | 5.3        |
> > > | GPTVQ (synth) | ~4    | 6.9       | 5.6        |
> > > | AQLM (synth)  | ~4    | 9.8       | 7.8        |
> > >
> > > At the 4-bit level, S-Quant achieves the best perplexity on both LLaMA2-7B and 13B.
> > >
> > > #### Around 3-bit
> > >
> > > | Method        | Bits | LLaMA2-7B | LLaMA2-13B |
> > > |---------------|------|-----------|------------|
> > > | S-Quant   | 2.7  | 6.5   | 5.8        |
> > > | QuIP#         | ~3    | 10.8      | 5.7    |
> > > | GPTVQ (synth) | ~3    | 11.8      | 6.4        |
> > > | AQLM (synth)  | ~3    | 15.6      | 13.0       |
> > >
> > > At the 3-bit level, S-Quant substantially outperforms all baselines on LLaMA2-7B and remains clearly stronger than GPTVQ-synth and AQLM-synth on LLaMA2-13B, while being comparable to QuIP#.
> > >
> > > #### Around 2-bit
> > >
> > > | Method        | Bits | LLaMA2-7B | LLaMA2-13B |
> > > |---------------|------|-----------|------------|
> > > | S-Quant   | 2.1  | 8.5   | 7.9    |
> > > | QuIP#         | ~2    | 9.8       | 8.5        |
> > > | GPTVQ (synth) | ~2    | 11.2      | 9.4        |
> > > | AQLM (synth)  | ~2    | inf       | inf        |
> > >
> > > At the 2-bit level, S-Quant remains stable across both model sizes, whereas QuIP# and GPTVQ-synth degrade noticeably, and AQLM-synth becomes completely unstable.
> > >
> > > ### Overall observation
> > >
> > > These results show that once the comparison is restricted to the data-free regime, methods that depend on calibration data deteriorate rapidly in the extreme low-bit setting, while our method continues to operate reliably.
> > >
> > > ### Conclusion and final version
> > >
> > > These results demonstrate that data-free quantization can remain effective even in the extreme low-bit regime, while data-dependent methods become unstable or degrade substantially once access to real calibration data is removed. In the final version, we will include these experiments, extend evaluation to larger models (e.g., 8B and 70B), and provide full implementation details. We thank the reviewer again for the valuable suggestions, which significantly improve our work.

---

### Official Review · Reviewer_opSP · 2026-03-15

**Soundness:** 3
**Presentation:** 2
**Significance:** 3
**Originality:** 3
**Overall Recommendation:** 4
**Confidence:** 3

**Summary:**

The paper presents S‑Quant which is a post‑training, weight‑only compression/quantization technique that reduces memory overhead (and memory bandwidth durign inference).  It splits each weight tensor into fixed‑size blocks, assigns each block a seed, and uses LFSR  to deterministically generate basis tensors from that seed.  Each block is then approximated as a linear combination of these generated bases, with block‑specific coefficients (which are the key elements which will be stored). The method searches over candidate seeds per block and increases the number of bases (k) until a certain error threshold is met. They also propose an ASIC‑style accelerator where seeds index a LUT of LFSR states and weights are reconstructed on‑the‑fly to improve performance.

**Compliance With Llm Reviewing Policy:**

Affirmed.

**Final Justification:**

The authors addressed most of the concerns I had about the paper. Keeping my prior score.

**Key Questions For Authors:**

1. Same question as described in Weakness #1 - Equation 13 does nto show that we are storing the number of k itself. Is there any assumption on the number of k for a given group of G blocks?
2. How is G selected?
3. Layerwise behavior - Did you observe that some layers consistently require larger k? In MoE models it might be more insightful to see how this differs across experts. Maybe it will reduce the search overhead during quantization.
4. Can you provide some more details on how the GPU kernel would look like during the reconstruction? How does the pipelining work in terms of fetching the coefficients, reconstructing the matrix, and computation of the matmul. Ideally you would have to retune the kernel to fit this kind of computation to achieve any gains.

**Limitations:**

1. It is not clear if the hardware design is necessary to achieve gains - and if it is necessary then it might be difficult to advocate for the unit to be introduced in an already area constrained GPU cores.

**Strengths And Weaknesses:**

Strenghts:
1. Innovative idea to use LFSR to generate base tensors and then use them on the fly. This directly addresses the memory capacity and memory bandwidth problem in inferencing.
2. The adaptive number of blocks gives a high level insight on why this could translate to larger and complex models.
Weaknesses:
1. It is unclear what the metadata would look like with adaptive number of blocks. Equation 13 tells us the overall cost of storage but the number of k will be different for each block - so there must have been a k stored for each block. But if we are not storing then there might be an assumption about the same number of blocks being used over a group G. It is unclear either way.
2. It is also not clear why an ASIC is required to enable more performance. In GPUs, one could use teh shared memory/RF to store the temporary created base tensors. Some more discussion on how tiling and pipelining would work would also been beneficial.

---

> ### Author Rebuttal · Authors · 2026-03-26
>
> ### Q1
> Thank you for pointing this out. This is a very helpful observation. Since k is selected adaptively for each block in our method, storing only the k coefficients is not sufficient for the decoder to determine the effective number of coefficients for that block. We would like to clarify that, in our actual design and implementation, we have always stored an additional 8-bit length field for each block to represent k, and all experiments in the paper were conducted based on this implementation. We appreciate the reviewer for identifying this issue in the written formula.
> Since the range of k in our experiments does not exceed 63, 8 bits are sufficient to represent this length information. Therefore, more precisely, the storage for each block should include the seed length, k int8 coefficients, one 8-bit field for the k length, and the amortized FP16 scaling factor shared across every G blocks. In other words, Eq.~(13) in the current manuscript omits this metadata term and is therefore incomplete. We will correct the equation in the final revision and explicitly clarify this implementation detail.
>
> ### Q2
> The group size G is treated as a design variable in our configuration space, together with the block size B, the seed length S, and the explained-energy threshold R_th. It is selected through the same Bayesian optimization process used for the other hyperparameters. Intuitively, G controls the granularity of coefficient dequantization: a larger G amortizes the FP16 scaling factor over more blocks and reduces metadata overhead, while a smaller G provides finer-grained scaling and may better preserve reconstruction quality. Therefore, G is chosen to balance storage efficiency and reconstruction accuracy.
>
> ### Q3
> Thank you for this insightful suggestion. Our current results suggest that the optimal adaptive basis count k is not uniform across layers, and some layers indeed appear to require a larger k than others to maintain reconstruction quality. For MoE models, we also agree that different experts may exhibit distinct compression sensitivities, making expert-wise analysis a meaningful direction. Such patterns could potentially help reduce the search space and lower the quantization search overhead. We will include a discussion of this point in the final revision.
>
> ### Q4
> Thank you for this question. We would like to clarify that the current hardware implementation and performance analysis in this work are not targeted at a GPU kernel, but rather at a custom seed-compression-aware accelerator. In fact, the reconstruction flow is already described in Section 3.5 and Appendix B: compact seeds and coefficients are first fetched from memory and fed into the weight generator; the generator then uses the seed as an index to perform a single-cycle LUT lookup to retrieve the corresponding tensor representation, followed by on-the-fly weight reconstruction using parallel multiplier / adder units; the reconstructed weights are then streamed directly into the systolic array for subsequent MAC operations, without requiring intermediate buffering.

---

> > ### Author Rebuttal · Reviewer_opSP · 2026-04-08
> >
> > Thanks for your response. The rebuttal fully resolves my concerns.

---

### Decision · Program_Chairs · 2026-04-30

**Decision:**

Accept (regular)

**Comment:**

S-Quant proposes a genuinely novel weight quantization scheme that uses compact seeds to drive LFSR-based basis matrix generation, enabling on-the-fly weight reconstruction and achieving state-of-the-art 3 to 4 bit compression across 7B to 70B LLMs with a dedicated ASIC accelerator delivering 4x speedup. The core idea is creative and directly addresses the memory bandwidth bottleneck in memory-bound inference. The rebuttal satisfactorily clarified the metadata storage design and the role of the group size G, though some desire remains for a clearer GPU kernel discussion and layer-wise k analysis. We recommend acceptance and ask the authors to correct Equation 13 to include the k-length metadata term and add a layer-wise k distribution analysis in the camera-ready version.